# Cross-subject decoding of human neural data for speech Brain Computer Interfaces

## Abstract

Brain-to-text systems have recently achieved impressive performance when trained on single-participant data, but remain limited by uninvestigated cross-subject generalization. We present the first neural-to-phoneme decoder trained jointly on the two largest intracortical speech datasets (Willett et al. 2023; Card et al. 2024), introducing day- and dataset-specific affine transforms to align neural activity into a shared space. A hierarchical GRU decoder with intermediate CTC supervision and feedback connections further mitigates the conditional-independence assumption of standard CTC loss. Our model matches or outperforms within-subject baselines while being trained across participants, and adapts to unseen subjects using only a linear transform or brief fine-tuning. On an independent inner-speech dataset (Kunz et al. 2025), our approach demonstrate generalization, by training only subject day specific transforms. These results highlight cross-subject pretraining as a practical path toward scalable and clinically deployable speech BCIs.

## 1 Introduction

Language is the cornerstone of human communication, and the ability to speak underpins social interaction, autonomy, and quality of life. The loss of speech—whether due to amyotrophic lateral sclerosis (ALS), stroke, or traumatic brain injury—often leads to social isolation and psychological distress Kao & Chan (2024). For individuals with intact cortical representations of speech but impaired motor output, neural speech prostheses offer a path to restoring communication by decoding intended speech directly from neural activity. The overall idea is to bypass muscle controls that could be impaired and directly decode the intended language or speech content from the related neural activity. Decoding speech from the brain has been approached using both non-invasive and invasive neural recordings. Non-invasive methods such as fMRI and MEG have demonstrated that semantic or phonetic content can be decoded above chance Défossez et al. (2023); Tang et al. (2023), but their temporal resolution and latency remain limiting factors for real-time communication. In contrast, invasive techniques such as electrocorticography (ECoG) and intracortical microelectrode arrays provide high-bandwidth, high-temporal-resolution access to speech-related cortical activity, enabling real-time speech decoding systems with word error rates (WER) approaching practical usability thresholds Silva et al. (2024); Willett et al. (2023); Card et al. (2024). However, invasive recordings present unique challenges: (i) It is invasive! Requires dedicated neurosurgical intervention; the eligible population is therefore limited, and the number of electrodes that can be safely implanted in humans is strictly constrained. (ii) Data collection is logistically demanding and clinically constrained, often yielding datasets from just a handful of participants worldwide; (iii) recordings are heterogeneous, with electrode placement driven by clinical need rather than research standardization; and (iv) neural signals exhibit nonstationarity over time, with electrode impedance changes and neural plasticity causing substantial within-subject drift. These factors have led most prior work to adopt a single-subject training paradigm, limiting model generalizability and hindering systematic progress.

A series of seminal studies have demonstrated the feasibility of decoding speech, phonemes, or articulatory kinematics from invasive neural data. Moses et al. Moses et al. (2019; 2021) achieved real-time decoding of a 50-word vocabulary using neural network models combined with n-gram language models. Willett et al. Willett et al. (2021; 2023; 2024) extended this line of work with a CTC-based recurrent neural network capable of phoneme-level decoding from thousands of sen-

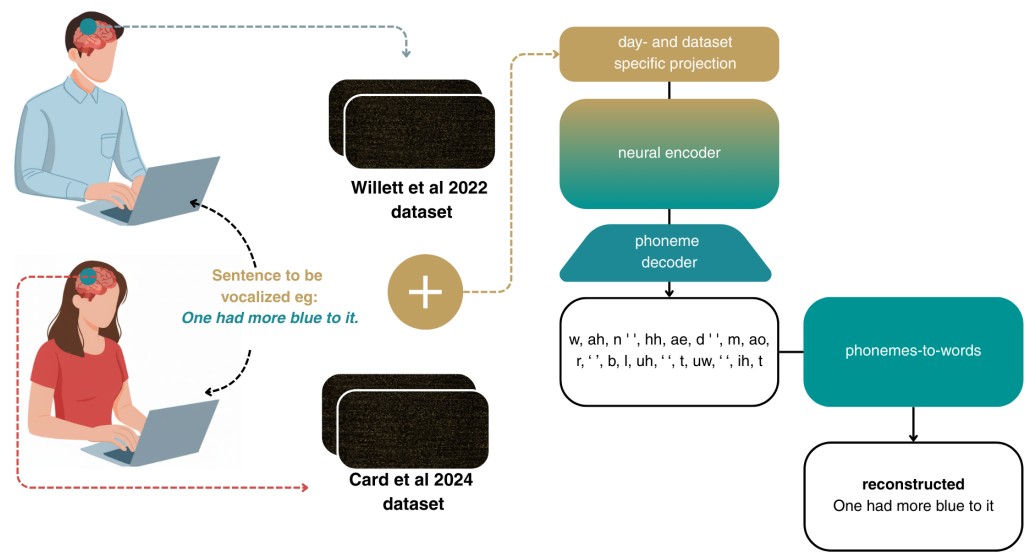

Figure 1: **Cross-subject neural speech decoding pipeline.** Neural data from Willett and Card participants are mapped into a shared space via day- and dataset-specific linear projections, encoded by a shared GRU-based model trained with hierarchical CTC, and decoded into open-vocabulary text through a phoneme-to-word module.

tences, achieving open-vocabulary WERs of 23.8% and communication rates exceeding 60 words per minute. More recently, Card et al. Card et al. (2024) demonstrated the most accurate conversational neuroprosthesis to date, achieving more than 90% accuracy across 125,000 words and supporting >30 WPM communication in spontaneous conversation. These studies confirm the viability of invasive brain-to-speech systems but leave open a crucial question: **can models trained on one participant generalize to others**?

The reliance on single-subject training is a major bottleneck for clinical translation. Each new user typically requires hours of supervised calibration data to achieve competitive performance, making deployment slow and resource-intensive. Yet there are reasons to believe that generalization across participants may be feasible (i) the cortical representation of speech is organized in a largely conserved topography across individuals Bouchard et al. (2013); Mugler et al. (2018); (ii) phoneme-related neural tuning properties have been shown to be reproducible across participants Stavisky et al. (2018); and (iii) within-subject signal drift over time can be as large as the variability observed across participants implanted in comparable regions. If cross-subject models could be trained, they might provide a better initialization for new users, reduce calibration requirements, and enable large-scale data aggregation—analogous to the role of pretraining in automatic speech recognition (ASR) Radford et al. (2022); Baevski et al. (2020). To address the challenge of single-participant dependence, we propose the first cross-subject neural-to-phoneme decoding model trained on invasive recordings from multiple participants implanted in distinct cortical regions. Rather than treating each participant as an isolated case, our approach leverages the shared structure of speech representations in the sensorimotor cortex to learn a unified decoder that can be adapted to new subjects. A couple of scientific challenges immediately arise here. The first is how to deal with neural variability over time. It is well known that neural representations drift quite a bit from day to day. To get an intuition for this, imagine you are asked to draw a circle on a piece of paper. It probably won't be a perfect circle, and if you try again you'll get another one—still a circle, but a slightly different one. Each attempt will vary in size, eccentricity, and asymmetry. Yet all these circles are similar enough that, with a simple affine transformation, we could align them quite well. To make them perfectly identical would require a nonlinear warp, but for alignment purposes a linear transform is usually good enough. Now imagine we also draw a circle. Our circle may be slightly larger or smaller than yours, maybe a bit more oval, but it is still recognizably a circle. With another affine transform, we could align our circle to yours too. This is exactly the intuition behind our approach: if two people can draw circles that can be aligned, maybe two participants' brains produce neural repre-

sentations of speech that can also be aligned into a shared space. We think this is feasible because all the people share an abstract concept of circle and they can somehow convert this "platonic" circle into an approximation in the real world. Language representations could behave similarly in neural representations.

In our work, we learn a subject-specific and day-specific linear transform to map each participant's neural data into a common space. The model then processes these aligned signals with its nonlinear causal decoder, focusing on capturing the complex phonetic and articulatory patterns that are shared across participants. This approach keeps the alignment step simple and interpretable while allowing the network to use its capacity where it matters most.

A second technical challenge is figuring out how to actually solve the decoding task. We start from neural data recorded during an attempted speech task, where each trial can have a different length and we have no direct alignment between neural time frames and words or phonemes. This means we cannot simply treat it as a supervised frame-by-frame classification problem. Recent work has borrowed heavily from ASR, using either autoregressive decoders or phoneme classifiers trained with the Connectionist Temporal Classification (CTC) loss. In the recent Brain2Text '24 Challenge Willett et al. (2024), a wide range of model architectures were tested—from GRUs to transformers and even state-space models. Interestingly, the simplest approach often performed the best: a relatively small GRU trained with CTC loss to predict phonemes, combined with a weighted finite-state transducer (WFST) to map phonemes to words. Most gains in final WER came not from radically new model designs but from ensembling or simply throwing more compute at training.

This result raises two possibilities: perhaps we do not yet have enough data to fully exploit modern architectures such as transformers or SSMs—or perhaps the GRU+CTC approach is simply the most data-efficient option for now. Still, we believe there is room for improvement, particularly because of a limitation inherent to the CTC objective: CTC assumes that each prediction is conditionally independent of the previous ones. This prevents the model from fully exploiting the joint probabilities between successive phonemes. Yet evidence from neuroscience suggests that the speech motor cortex does not just encode isolated phonemes but also their transitions—at least at the level of diphonemes Xu et al. (2024). Indeed, the first-place entry in the challenge achieved a small boost in phoneme error rate by adding an auxiliary diphoneme prediction head, though at the cost of a much larger class space (quadratic in the number of phonemes) and a more complex training process Li et al. (2024).

Autoregressive transformer models trained with cross-entropy could, in theory, capture these dependencies. However, in practice, they still lag behind CTC in this domain, and their training tends to be unstable. To get the best of both worlds, we build on the proven GRU architecture but propose a novel hierarchical loss function. In our approach, phoneme predictions are generated at multiple depths of the network and then fed back into the subsequent recurrent layers. This provides the model with explicit information about its own phoneme hypotheses at earlier stages, allowing it to refine predictions in a way that partially recovers the conditional modeling power of autoregressive approaches—while keeping training as simple and robust as standard CTC. (See Figure 1 for a scheme of the decoding pipeline).

We evaluate this approach by aggregating all the largest publicly available speech BCI datasets Willett et al. (2024); Card et al. (2024); Kunz et al. (2025) and comparing cross-subject models to state-of-the-art within-subject baselines. Our results show that cross-subject training is not only feasible but also does not degrade performance relative to single-subject models. Moreover, we find that models trained in this way can be rapidly adapted to new participants with minimal fine-tuning data—achieving competitive phoneme error rates. Taken together, our findings demonstrate that cross-subject generalization is a realistic and promising path forward for neural speech decoding. By pooling data across participants, we move closer to the vision of foundation models for BCIs: models that can be trained once on large, diverse datasets and then deployed with minimal retraining for new users. Such a paradigm has transformed fields like natural language processing and speech recognition, where pretrained models dramatically lower the data requirements for downstream tasks. In the context of neural decoding, this could mean reducing the calibration burden for new patients accelerating clinical translation.

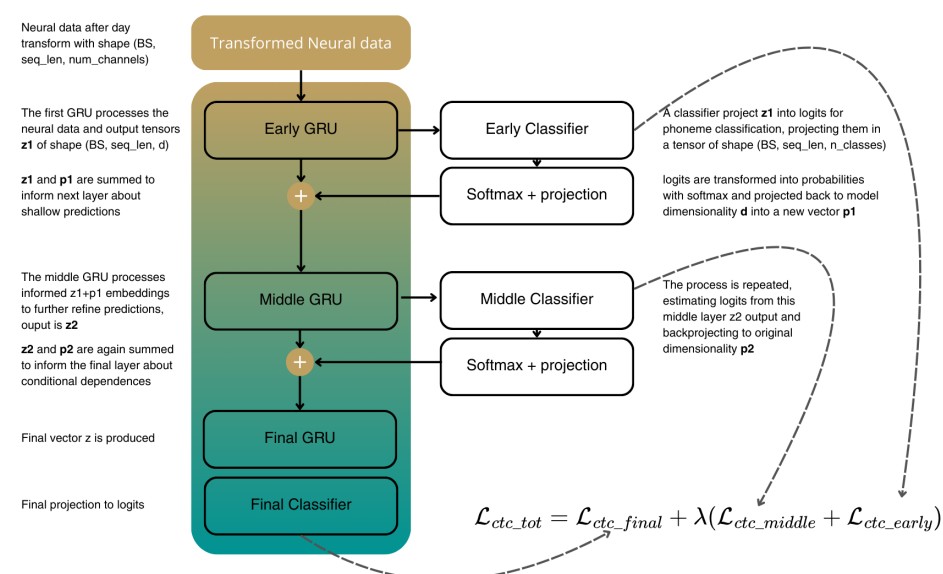

Figure 2: **Hierarchical GRU decoder with feedback.** Neural features are processed by three stacked GRUs. Each of the first two GRUs produces phoneme predictions ($p_1$, $p_2$) that are projected back and added to their hidden states, guiding deeper layers. Training uses a hierarchical CTC loss, combining early, middle, and final predictions.

## 2 METHODS

### 2.1 DATA

We conduct our experiments by aggregating all publicly available sources of speech decoding datasets from the BrainGate2 initiative. The first dataset ("Willett") Willett et al. (2023) is from subject T12. Data were collected from a single participant with ALS and anarthria implanted with four 64-channel Utah microelectrode arrays (256 channels total), targeting two key regions of the speech motor network: the ventral premotor cortex (Brodmann area 6v) and Broca's area (area 44). Electrode placement was guided by subject-specific fMRI activation maps and structural parcellation from the Human Connectome Project. During the task, the participant attempted to speak prompted sentences in an instructed-delay paradigm. Raw neural signals were bandpass filtered in the ultra-high gamma range (250-5000 Hz). Threshold crossings were detected using a fixed threshold of $-4.5$ RMS, and spike counts were binned in non-overlapping 20 ms windows. In parallel, spike band power was computed as the mean squared signal in each 20 ms bin. These two features (counts and power) were concatenated to form a neural feature vector per channel per time bin. The dataset spans 24 days of recordings, totaling approximately 9,000 trials. We follow the official train-test-competition split provided by the authors, using temporally separated evaluation blocks to test generalization across time and relying only on the premotor cortex electrodes since ones in the Broca region where no or just little informative. The second dataset ("Card") is from subject T15 Card et al. (2024). This dataset comprises recordings from a participant with ALS and severe dysarthria implanted with four 64-channel Utah arrays (256 channels) in the left ventral precentral gyrus. Neural features consist of spike counts and spike-band power, preprocessed similarly to Willett et al. The dataset includes >8 months of recordings across 84 sessions and 45 different days. Again we relied on the original train/test/competition split provided by authors. We first trained our models on the concatenation of the Willett et al. and Card et al. datasets, which together constitute the largest publicly available collection of intracortical neural recordings during attempted speech. Merging these datasets allowed us to maximize data diversity across days, sessions, and cortical coverage, providing a strong foundation for learning robust neural-to-phoneme mappings.

To probe the limits of cross-subject and cross-task generalization, we further evaluated our models on the dataset introduced by Kunz et al. Kunz et al. (2025). This dataset is unique in that it focuses primarily on inner speech rather than overt production, offering a qualitatively different neural

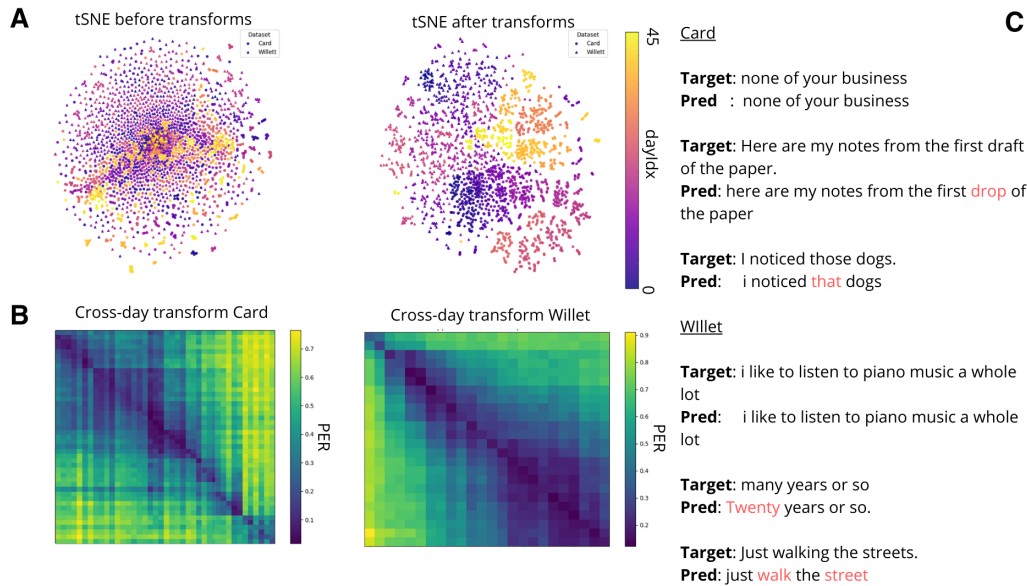

Figure 3: **Effect of day transforms and example outputs. A:** t-SNE of neural embeddings before/after day transforms, showing reduced day clustering after alignment. **B:** Cross-day transform swapping on Card and Willett: lowest PERs lie on the diagonal, but off-diagonals remain reasonable, indicating shared structure across days. **C:** Example sentence-level predictions, with most errors being minor word substitutions.

regime. Recordings were obtained from four participants (T12, T15, T16, T17) implanted with Utah arrays in ventral motor and premotor speech areas. Notably, T12 and T15 are the same participants featured in the Willett and Card datasets but were re-recorded several months later, making these data ideal for testing long-term stability and generalization under neural signal drift. Participants T16 and T17 are entirely new subjects, providing a clean setting to evaluate out-of-subject generalization. The corpus spans several experimental categories: isolated verbal behaviors, which include listening, reading, mouthed speech (silent articulation without phonation), attempted speech (covert articulation with intended production), and imagined speech (purely internal generation); sentence datasets, where participants attempted or imagined speaking either small controlled vocabularies (50 words) or much larger sets (up to 125k words); as well as additional paradigms such as interleaved verbal behaviors, conjunctive counting, and sequence recall. For this study, we restricted to speech-motor–related electrodes From the isolated verbal behaviors category we included only the attempted and mouthed conditions, while from the sentence data we used attempted and imagined trials. This selection emphasizes speech production and imagination processes while excluding purely perceptual conditions (listening, reading). After preprocessing and trial selection, the resulting dataset comprised 836 trials from t12 (5 sessions), 1,040 trials from t15 (9 sessions), 224 trials from t16 (2 sessions), and 320 trials from t17 (2 sessions), for a total of 2,420 trials. Features were standardized and padded to a 512-dimensional representation for downstream modeling.

## 2.2 Neural Encoder and Hierarchical GRU Decoder

**Day- and Subject-Specific Transformation.** Before entering the encoder, neural features are first passed through a subject- and day-specific affine projection to compensate for variability across recording sessions and participants. For each subject $s$ and recording day $d$, we learn a linear transform $\tilde{\mathbf{x}}_t^{(d,s)} = \mathbf{W}_{d,s}\mathbf{x}_t + \mathbf{b}_{d,s}$, where $\mathbf{x}_t \in \mathbb{R}^C$ is the neural feature vector at time $t$ (with $C$ channels), and $\mathbf{W}_{d,s} \in \mathbb{R}^{C \times C}, \mathbf{b}_{d,s} \in \mathbb{R}^C$ are trainable projection weights and biases. This projection aligns neural data into a shared latent space, mitigating electrode drift and subject-specific scaling effects.

**Model Architecture.** The transformed data $\tilde{\mathbf{X}} \in \mathbb{R}^{B \times T \times C}$ are processed by a three-block hierarchical GRU decoder (Figure 2). The first two blocks contain two bidirectional GRU layers each,

while the final block contains a single GRU layer. We denote the hidden dimensionality by $d$ and the number of phoneme classes (including the CTC blank) by $N$.

**Early GRU Block.** The early GRU block computes

$$\mathbf{z}_1 = \mathrm{GRU}^{(2)}_{\mathrm{early}}(\tilde{\mathbf{X}}), \qquad \mathbf{z}_1 \in \mathbb{R}^{B \times T \times d}. \tag{1}$$

An auxiliary classifier projects $\mathbf{z}_1$ into phoneme logits:

$$\boldsymbol{\ell}_1 = \mathbf{W}_{\mathrm{early}} \mathbf{z}_1 + \mathbf{b}_{\mathrm{early}}, \qquad \boldsymbol{\ell}_1 \in \mathbb{R}^{B \times T \times N}. \tag{2}$$

Softmax produces class probabilities

$$\mathbf{p}_1 = \mathrm{Softmax}(\boldsymbol{\ell}_1), \qquad \mathbf{p}_1 \in [0,1]^{B \times T \times N}, \tag{3}$$

which are projected back to the hidden dimension:

$$\hat{\mathbf{p}}_1 = \mathbf{W}_{\mathrm{proj},1} \mathbf{p}_1 + \mathbf{b}_{\mathrm{proj},1}, \qquad \hat{\mathbf{p}}_1 \in \mathbb{R}^{B \times T \times d}. \tag{4}$$

The feedback signal is summed with the original hidden states to form the input to the next block: $\mathbf{h}_1 = \mathbf{z}_1 + \hat{\mathbf{p}}_1$.

**Middle GRU Block.** Similarly, the middle block refines the representation: $\mathbf{z}_2 = \mathrm{GRU}^{(2)}_{\mathrm{middle}}(\mathbf{h}_1)$, with another auxiliary classifier producing $\boldsymbol{\ell}_2 = \mathbf{W}_{\mathrm{middle}} \mathbf{z}_2 + \mathbf{b}_{\mathrm{middle}}$, $\mathbf{p}_2 = \mathrm{Softmax}(\boldsymbol{\ell}_2)$, $\hat{\mathbf{p}}_2 = \mathbf{W}_{\mathrm{proj},2} \mathbf{p}_2 + \mathbf{b}_{\mathrm{proj},2}$. We again sum the feedback signal to inform the final layer: $\mathbf{h}_2 = \mathbf{z}_2 + \hat{\mathbf{p}}_2$.

**Final GRU Block.** The final block consists of a single GRU layer: $\mathbf{z}_3 = \mathrm{GRU}^{(1)}_{\mathrm{final}}(\mathbf{h}_2)$, followed by the final projection to phoneme logits: $\boldsymbol{\ell}_3 = \mathbf{W}_{\mathrm{final}} \mathbf{z}_3 + \mathbf{b}_{\mathrm{final}}$.

**Hierarchical CTC Loss.** Because frame-level phoneme alignment is not available, we train with the Connectionist Temporal Classification (CTC) loss Graves et al. (2006), which marginalizes over all possible alignments:

$$\mathcal{L}_{\mathrm{CTC}}(\boldsymbol{\ell}, \mathbf{y}) = -\log \sum_{\pi \in \mathcal{B}^{-1}(\mathbf{y})} \prod_{t=1}^{T} P(\pi_t \mid \boldsymbol{\ell}_t), \tag{5}$$

where $\pi$ is a valid alignment path and $\mathcal{B}$ is the collapse operator that removes blanks and repeats. A known limitation of CTC is its *conditional independence assumption*: predictions $P(\pi_t)$ are modeled independently across time steps. Our hierarchical decoder partially mitigates this limitation by feeding back layer-wise phoneme probabilities into deeper GRU blocks, allowing later representations to be informed by earlier hypotheses.

The total training loss combines CTC terms from all three layers:

$$\mathcal{L}_{\mathrm{CTC,total}} = \mathcal{L}_{\mathrm{CTC}}(\boldsymbol{\ell}_3, \mathbf{y}) + \lambda \Big[ \mathcal{L}_{\mathrm{CTC}}(\boldsymbol{\ell}_2, \mathbf{y}) + \mathcal{L}_{\mathrm{CTC}}(\boldsymbol{\ell}_1, \mathbf{y}) \Big], \tag{6}$$

where $\lambda \in [0,1]$ balances the auxiliary supervision terms.

**Training Details.** Models are trained jointly on the Willett and Card datasets for **120k steps** with a batch size of **64** using the Adam optimizer. The learning rate is linearly warmed up from 0 to $5 \times 10^{-3}$ over the first **1k steps**, followed by cosine decay to $1 \times 10^{-4}$ at step **120k**. We apply a weight decay of $1 \times 10^{-5}$ and use mixed-precision training with gradient accumulation to maximize GPU memory efficiency. Gaussian noise and small per-channel offsets are applied as data augmentation to improve robustness. Hyperparameters where chosen to be identical to original Card baseline expect for model dimensionality set at $d = 2048$ to let the model have more capacity to deal with larger neural subspace. $\lambda$ was set a 0.3 empirically. Further hyperparameter exploration could boost the performance and was left as future work.

Table 1: **Main results on the Willett and Card datasets.** We report phoneme error rate (PER), word error rate (WER) on our evaluation set, and the official Brain2Text'24 and Brain-to-text 2025 challenges WER for comparison.

| Model | Training Set | Test Set | PER (%) | WER (%) / Comp. WER (%) |
|---|---|---|---|---|
| Willett baseline | Willett | Willett | 19.7 | 17.4 / 11.06 |
| Card baseline | Card | Card | 10.2 | 7.34 / 6.70 |
| Ours (plain CTC) | Willett+Card | Willett | 17.6 | 14.54 / 10.9 |
| Ours (plain CTC) | Willett+Card | Card | 9.6 | 7.57 / 6.39 |
| **Ours (hierarchical CTC)** | Willett+Card | Willett | **16.1** | **14.54 / 10.3** |
| **Ours (hierarchical CTC)** | Willett+Card | Card | **9.1** | **6.67 / 5.9** |

## 2.3 PHONEME-TO-WORD DECODING AND EVALUATION

The final stage of our pipeline transforms the decoded phoneme sequences into meaningful sentences. In line with previous works, we relied on a classical approach based on weighted finite-state transducers (WFSTs), which integrate phoneme posteriors with lexicon and language model constraints. Here, the output lattice from the neural decoder is composed with a pronunciation lexicon and a 5-gram language model, and the most likely sequence is recovered using beam search. Optionally, the resulting hypotheses can be rescored using a larger pretrained language model such as OPT (see Willett et al. (2023) for details). WFST-based decoding remains a strong baseline, offering reliable performance, but is computationally expensive, memory-intensive, and inherently limited to a fixed context window of a few words. We assess performance using two widely adopted metrics: Phoneme Error Rate (PER) and Word Error Rate (WER). PER is computed as the normalized edit distance between the predicted and reference phoneme sequences, counting substitutions, insertions, and deletions. WER is computed over the final text output, enabling a direct comparison between WFST-based and neural decoding approaches at the sentence level. All experiments are conducted on the official held-out test sets, with WER also reported on the competition sets (where PER is not computable since we don't have access to ground truth labels and WER is returned by online platforms).

## 2.4 CROSS-SUBJECT GENERALIZATION

To evaluate whether our model can generalize beyond the participants used for training, we selected the best-performing model trained jointly on the Willett and Card datasets and froze all its parameters except for the subject/day-specific input transformations. We then introduced a new linear transform for each participant in the Kunz et al. dataset and optimized only these transforms on the available data. This procedure effectively performs a lightweight alignment of each new subject's neural feature space to the shared latent space learned during pretraining. The number of trainable parameters per subject is small ($\mathcal{O}(C^2 + C)$), making this adaptation fast and data-efficient. This experiment enables us to assess whether a model pretrained on overt speech from multiple participants can quickly adapt to new participants performing a different task (inner speech) with minimal data.

## 2.5 ANALYSIS OF DAY-SPECIFIC TRANSFORMS

To better understand the contribution of the day-specific affine projections, we analyzed the learned transforms $\{W_d, b_d\}$ both qualitatively and quantitatively. We first visualized neural embeddings before and after transformation using t-SNE, averaging neural activity per trial to highlight global day-level structure. After transformation, embeddings from different days became significantly more clustered, suggesting that the transforms successfully normalize session-to-session variability. Finally, we performed a transform swapping experiment, in which each day's data was processed using every other day's transform, and measured the resulting PER. This analysis quantifies the similarity between days and evaluates how sensitive decoding performance is to transform mismatch.

## 3 RESULTS

### 3.1 JOINT TRAINING ACROSS SUBJECTS IMPROVES PERFORMANCE

Table 1 summarizes the performance of our models trained on the Willett and Card datasets, reporting phoneme error rate (PER) and word error rate (WER) on each dataset's held-out evaluation

Table 2: **Cross-subject generalization to the Kunz et al. dataset.** We report Phoneme Error Rate (PER, %) when evaluating the best cross-subject model on four participants (T12–T17) under three adaptation regimes: training only subject-specific linear transforms, fine-tuning the entire model, and training from scratch on the target data.

| Generalization Target | Training Only Linear | Fine-Tuning Whole Model | From Scratch |
|---|---|---|---|
| T12 (Kunz) | 30.2 | 21.3 | 11.8 |
| T15 (Kunz) | 28.8 | 26.3 | 26.1 |
| T16 (Kunz) | 41.1 | 26.1 | 40.9 |
| T17 (Kunz) | 58.9 | 53.3 | 30.6 |

blocks, as well as the official competitions WER for reference. Training a single model jointly on both datasets yields comparable or better performance than models trained separately on each dataset and this is our first and main result: cross-subject training is feasible. Specifically, our joint model with a plain CTC loss improves Willett PER from 19.7% to 17.6% and WER from 17.4% to 14.5%, while the performance of single-subject baselines on Card at matched by our model. This demonstrates that cross-subject training is feasible and does not degrade subject-specific performance. Furthermore, our proposed **hierarchical CTC decoder** exhibit a modereate improvement in performance across the board. On Willett, PER is reduced to 16.1%, with a relative WER reduction compared to the plain CTC model. On Card, the improvements are even more pronounced, reaching 9.1% PER and 6.67% WER, outperforming the single-subject baseline. These results confirm that (i) cross-subject training is not only possible but beneficial, and (ii) our hierarchical CTC design partially mitigates some of the conditional independence limitations of the standard CTC objective and improves performances.

### 3.2 Effect of Day-Specific Transforms

Figure 3A shows a t-SNE visualization of trial-averaged neural embeddings before and after applying the day-specific affine projections. Prior to transformation, the embeddings form a diffuse cloud with no discernible organization by day. After applying the transforms, a clear structure emerges: trials cluster consistently by subject and day, creating an organized space. This suggests that the day-specific transforms linearly re-center/rotate the data into a shared space that exposes task-relevant geometry. Figure 3B presents cross-day transform swapping experiments, where each day's transform was applied to every other day's data and the resulting PER was measured. In both datasets, the diagonal of the matrix (correct transform applied) yields the lowest PER, as expected, but many off-diagonal entries also achieve reasonable performance, suggesting that the transforms share structure and are not simply overfitting to individual days. This provides evidence that the learned projections capture generalizable session-invariant mappings and could be an hint that maybe less than one transform per day is actually needed. Finally, Figure 3C presents representative sentence-level predictions drawn from the held-out test set. Examples correspond to the 25th, 50th, and 90th percentiles of WER for each dataset, illustrating performance across easy, median, and challenging cases. In most examples, the two-stage decoding pipeline recovers the exact target sentence or a very close paraphrase, with remaining errors typically consisting of minor word substitutions or function-word variations rather than gross semantic failures.

### 3.3 Cross-Subject Generalization to Kunz et al. Participants

Table 2 reports phoneme error rates (PER) for four participants from the Kunz et al. dataset (T12–T17) under three adaptation regimes. Remarkably, simply training the subject-specific linear transforms on the new data already yields a substantial reduction in error compared to chance level (PER=100%), demonstrating that much of the variability between participants can be compensated by a lightweight affine re-alignment. Fine-tuning the entire model for a small number of steps (5k) further improves performance, reducing PER by an additional 20–40% relative to the linear-only adaptation. Interestingly, for some participants, training the model entirely from scratch can achieve even lower PER, likely due to the simplified nature of the Kunz dataset, where the majority of trials consist of a small set of seven single words repeated many times and only a fraction of full-sentence trials (for example t16 and t17 only have the 7 words trials, that are easier to overfit for the model). This setting favors models trained from scratch, which can specialize fully to the limited vocabu-

lary. Nonetheless, the cross-subject pretrained model with fine-tuning achieves competitive results with a fraction of the training time, highlighting its potential for rapid deployment in low-resource scenarios.

## 4 DISCUSSION AND CONCLUSION

Our study demonstrates that training neural speech decoders across multiple participants is not only feasible but beneficial: a single model trained on the concatenation of the Willett and Card datasets matches or outperforms single-subject baselines and generalizes well to new data. By introducing a hierarchical CTC loss with feedback connections, we mitigate some of the conditional-independence limitations of the standard CTC objective while preserving its stability and efficiency. Together, these results indicate that cross-subject pretraining can serve as a powerful strategy to bootstrap neural speech BCIs, enabling rapid adaptation to new participants with minimal calibration. A central contribution of this work is the explicit modeling of session and subject variability via learned day-specific affine transforms. Our analyses reveal that these transforms are not merely per-channel re-scaling layers: they perform meaningful linear re-alignments that re-align day-specific variability and highlight task-relevant structure in the neural data, as evidenced by t-SNE visualizations and transform swapping experiments. This is consistent with the hypothesis that neural manifolds are stable up to a low-dimensional linear transforms, and that much of the day-to-day drift can be corrected with simple affine mappings. Such transforms may provide a general mechanism for neural domain adaptation in BCIs, reducing the need for re-training large models when electrodes shift or signal statistics drift over time. Despite the improvements introduced by the hierarchical CTC decoder, our approach still inherits the conditional independence limitation of CTC at the frame level, which may prevent the model from fully capturing sequential dependencies in the phoneme stream. While the feedback connections partly reintroduce conditional information, there remains a gap between this approach and fully autoregressive sequence models. Downstream language models, whether WFST-based or neural, play an essential role in correcting phoneme-level errors and producing fluent sentences. As shown in our percentile-based qualitative analysis, most residual errors are attributable to phoneme-to-word reconstruction rather than neural-to-phoneme decoding, suggesting that future gains may come from more powerful phoneme-to-word models, ensembling, and context-aware decoding strategies. As neural decoding technology approaches practical usability, its ethical implications become increasingly critical. High-performance neural decoders could, in principle, extract unintended or private mental content. Recent work has shown that it is possible to decode selective attention and even imagined inner speech from cortical activity Tang et al. (2023); Kunz et al. (2025), raising concerns about privacy and consent in neural data use. We stress that decoding should be performed only with the participant's explicit intention and consent, in line with the development of intent BCIs. This may include requiring a separate neural signal for voluntary activation or the use of secure mental passwords as demonstrated in Kuntz et al. Kunz et al. (2025). Building safeguards into BCI systems is not just a technical challenge but an ethical necessity. Overall, our findings suggest several promising avenues for future work. First, scaling cross-subject training to larger, more diverse datasets could yield general-purpose foundation models for speech BCIs, analogous to Whisper Radford et al. (2022) or wav2vec Baevski et al. (2020) in ASR. Such models could be fine-tuned with a few minutes or hours of data to personalize decoding to a new participant. Second, integrating information from higher-order language and semantic regions may enable models to go beyond phoneme-level decoding and recover intended meaning, opening the door to concept-level or semantic BCIs. Architecturally, mixture-of-experts (MoE) models could be used to automatically select specialized experts conditioned on day or subject embeddings, offering a flexible way to handle variability across sessions. Finally, improving language modeling remains a key bottleneck: better phoneme-to-word models, rescoring strategies, and even in-context learning could further reduce WER and bring performance closer to natural conversation rates. In summary, this work shows that cross-subject training combined with lightweight subject-specific adaptation is a viable path toward scalable and robust neural speech BCIs. By aligning neural manifolds across participants and introducing a hierarchical CTC objective, we achieve performance competitive with single-subject systems while greatly improving data efficiency. These results lay the groundwork for future neural speech decoders that are not just accurate, but adaptable, efficient, and ethically deployable in real-world settings.

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

# A    STATEMENTS

## ETHICS STATEMENT

This work exclusively uses publicly available, de-identified datasets released by their original authors with explicit Institutional Review Board (IRB) approval and participant informed consent as described in the source publications (Willett et al., 2023; Card et al., 2024; Kunz et al., 2025). We did not collect any new human data, nor did we access identifiable or restricted information beyond the public releases. All data handling adheres to the licenses and usage terms specified by the dataset providers, and no attempt was made to re-identify individuals or to infer private attributes unrelated to the stated research purposes.

Neural speech decoding raises important privacy and autonomy considerations. Consistent with the intent-BCI paradigm, models should be deployed only with explicit user consent and with safeguards that prevent unintended decoding (e.g., explicit activation signals, user-controlled on/off mechanisms, and audit logs). We encourage practitioners to follow best practices for secure data storage, access control, and transparent communication of model capabilities and limitations. Where clinically applicable, systems should be developed in partnership with participants, clinicians, and ethics boards to ensure that benefits and risks are carefully balanced.

## REPRODUCIBILITY STATEMENT

We aim for full reproducibility. The Methods section specifies all architectural components, training schedules, optimization details, and evaluation protocols. We provide a self-contained `.zip` archive in the supplementary material that includes: (i) scripts to download/prepare the publicly available datasets used in this study (with checksums and expected directory structures), (ii) training and evaluation code, (iii) configuration files with all hyperparameters and random seeds for each experiment, (iv) instructions to reproduce Tables 1–2 and Figures 1–3.

Upon publication, we will release the same code publicly under a permissive license, together with frozen configuration files and (where dataset terms allow) pretrained checkpoints required to reproduce the reported results. Any deviations from defaults needed to match the paper's numbers are documented in the provided `README` and per-experiment config files.

# B    DATASET DETAILS

## B.1    DATASET STATISTICS

Table 3 summarizes the size and split of the three datasets used in this study: the Card dataset Card et al. (2024), the Willett dataset Willett et al. (2023), and the Kunz et al. dataset Kunz et al. (2025). For Card and Willett, we report the number of training, held-out test, and competition trials provided in the Brain-to-Text challenges. For Kunz et al., we report the number of trials after preprocessing and trial selection, split 80/20% into training and validation partitions.

Table 3: **Dataset splits and statistics used in this work.**

| Dataset | Training Trials | Test Trials | Competition Trials |
|---|---|---|---|
| Card (Brain-to-Text '25) | 8,072 | 1,426 | 1,450 |
| Willett (Brain-to-Text '24) | 8,800 | 880 | 1,200 |
| Kunz et al. (Inner Speech) | T12: 836, T15: 1,040, T16: 224, T17: 320 (total: 2,420) | | |

For the Kunz dataset, features were standardized and zero-padded to a 512-dimensional representation before being passed to the model. This ensured consistent input dimensionality across subjects and sessions.

### B.2 Participant and Recording Details

Table 4 compares the participants and experimental setups between the Brain-to-Text '24 (Willett) and Brain-to-Text '25 (Card) challenges.

Table 4: **Participant and experimental details for the Brain-to-Text '24 and '25 datasets.**

|  | **Brain-to-Text '24 (Willett)** | **Brain-to-Text '25 (Card)** |
|---|---|---|
| **Participant(s)** | T12, implanted with four 64-channel Utah arrays (128 electrodes in speech motor cortex, 128 in inferior frontal gyrus) | T15, implanted with four 64-channel Utah arrays (256 electrodes in speech motor cortex) |
| **Recording Period** | 25 sessions spanning 4 months | 45 sessions spanning 20 months |
| **Number of Sentences** | 12,100 | 10,948 |
| **Sentence Corpus** | Switchboard | 50-word vocabulary, Switchboard, OpenWebText2, Harvard sentences, custom high-frequency word sentences, random word sentences |
| **Speaking Strategy** | Attempted vocalized speech | Attempted vocalized or attempted silent speech |
| **Speaking Rate** | $\sim$62 words per minute | $\sim$30 wpm (vocalized) or $\sim$50 wpm (silent) |

### B.3 Task Design for Kunz et al. Dataset

Participants T12, T15, T16, and T17 performed perceptual tasks including listening and silent reading, followed by attempted or imagined speech production. Trials were drawn from a mixture of isolated words and full sentences, with the majority of trials consisting of a repeated set of seven single words. This design provides a valuable testbed for assessing cross-subject generalization under reduced linguistic complexity and weaker neural responses (inner speech).

## C Analysis of Subject and Day transforms

To better understand how the learned day-specific affine projections $\tilde{x} = W_d x + b_d$ behave across recording sessions and participants, we performed a quantitative analysis of the matrices $W_d$ and biases $b_d$. Each affine transform aims to compensate for day-to-day subject-wise variability in neural signal statistics, aligning the raw neural activity into a common latent space before decoding. We computed several matrix- and vector-based metrics that characterize the geometry, structure, and magnitude of each transform. Below we define these metrics, outline their interpretation, and discuss the patterns observed across datasets (Card and Willett). Figures 4–10 summarize the evolution of each metric across days, jointly visualizing the per-day Phoneme Error Rate (PER) obtained by the model after applying the corresponding transform.

### C.1 Definitions and Interpretation of Metrics

For each recording day $d$, with learned transform $W_d \in \mathbb{R}^{C \times C}$ and bias $b_d \in \mathbb{R}^C$, we compute:

**Frobenius distance to identity.**

$$\text{fro\_to\_I}(W_d) = \|W_d - I\|_F.$$

Measures how far $W_d$ deviates from the identity. Small values indicate minimal transformation; large values indicate substantial scaling or rotation.

**Condition number.**

$$\text{cond}(W_d) = \kappa(W_d) = \frac{\sigma_{\max}(W_d)}{\sigma_{\min}(W_d)}.$$

Large values correspond to anisotropic stretching that scale some channels more than others.

**Log absolute determinant.**

$$\text{logdet\_abs}(W_d) = \log |\det W_d|.$$

Reflects global volume change induced by the transform. Values near zero imply approximately volume-preserving mappings; strongly negative values indicate contraction.

**Orthogonality gap.**

$$\text{orth\_gap}(W_d) = \|W_d^\top W_d - I\|_F.$$

Quantifies deviation from an orthogonal transformation. Orthogonal matrices preserve angles and norms, so larger values indicate distortion of geometric structure. A value of zero would indicate that transforms are just rotation, while larger values suggest that shearing is also happening.

**Diagonal ratio.**

$$\text{diag\_ratio}(W_d) = \frac{\|\text{diag}(W_d)\|_F^2}{\|W_d\|_F^2}.$$

Values close to 1 imply low cross-channel mixing; lower values indicate greater off-diagonal structure.

**Off-diagonal energy.**

$$\text{offdiag\_energy}(W_d) = \sum_{i \neq j} W_{d,ij}^2.$$

Direct measure of how much channels are mixed by the transform.

**Spectral entropy.**

$$\text{spec\_entropy}(W_d) = -\sum_i p_i \log p_i, \qquad p_i = \frac{\sigma_i(W_d)}{\sum_j \sigma_j(W_d)}.$$

Higher entropy corresponds to more uniform distribution of singular values.

**$\ell_2$ bias magnitude.**

$$\text{bias\_l2}(b_d) = \|b_d\|_2.$$

Measures global shift applied to neural features.

In all figures, the left axis tracks the metric of interest and the right axis shows the PER achieved when decoding data from the same day.

C.2   SUMMARY OF OBSERVED PATTERNS

Across all metrics, we observe systematic differences between the two datasets, reflecting the distinct recording properties of the Card and Willett datasets.

**(1) Card shows smaller and more stable transforms.**   For metrics such as **fro_to_I**, **orth_gap**, **logdet_abs**, and **offdiag_energy**, Card consistently exhibits smaller magnitudes and narrower day-to-day variation. This indicates that Card's neural feature distribution is more stable across sessions, requiring only mild linear corrections.

**(2) Willett shows larger drift and stronger cross-channel mixing.**   In Willett, **cond**, **orth_gap**, **offdiag_energy**, and **bias_l2** attain larger values, particularly in early and late sessions. This reflects more pronounced day-to-day variability, greater signal drift, and increased need for alignment. Notably, early Willett days include extremely high condition numbers, suggesting highly anisotropic neural scaling.

**(3) Diagonal structure remains dominant.** For both datasets, **diag_ratio** remains near 0.9–1.0, indicating that most transforms preserve approximate channel independence and rely only modestly on cross-channel mixing. This supports our claim that the day-to-day drift is largely linear and can be corrected with simple affine operations.

**(4) Relationship with decoding accuracy.** We compute Pearson correlations between each metric and the corresponding PER for that day (annotated in each panel). For most metrics, especially in Willett, the correlation magnitude is moderate ($|r| \approx 0.4$–$0.6$), suggesting that days requiring stronger linear alignment tend to be harder to decode. This provides additional evidence that the day transforms capture true physical variability in neural signals rather than overfitting noise.

**(5) PER consistently decreases once transforms stabilize.** In both datasets, PER tends to fall when the geometric metrics stabilize (e.g., mid-to-late Card days; middle Willett days), indicating that the affine alignment effectively normalizes the input space for decoding.

## C.3 FIGURES

**Bias magnitude across days.** The evolution of $\|b_d\|_2$ and PER is shown in Figure 4. Card shows small, stable biases, while Willett exhibits larger drifts, particularly at the beginning and end of the recording period.

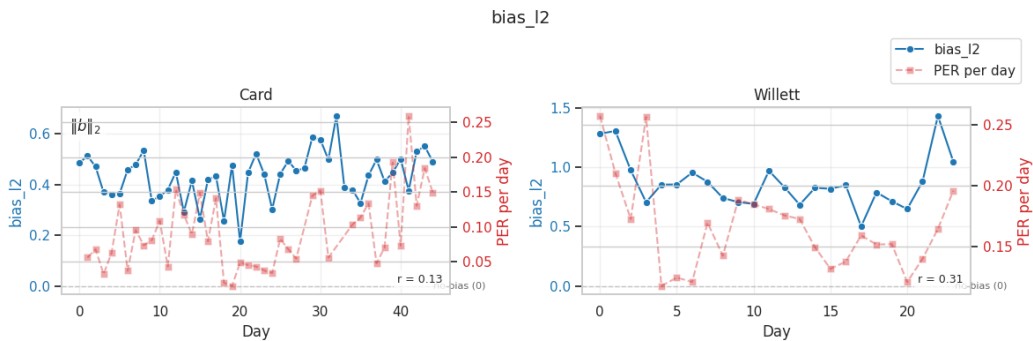

Figure 4: Day-wise evolution of $\|b_d\|_2$ (left axis) and PER (right axis).

**Condition number.** Figure 5 reveals substantial differences across datasets: Card maintains moderate condition numbers (4–12), whereas Willett shows extreme early-session instability ($\kappa(W_d) > 1000$), followed by stabilization.

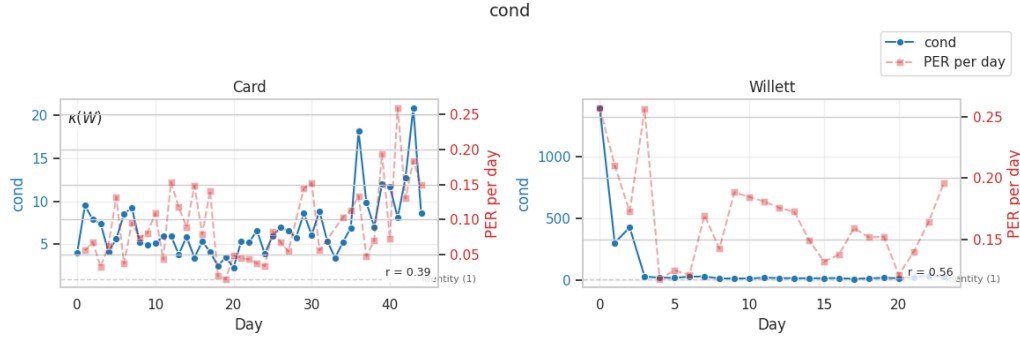

Figure 5: Day-wise evolution of $\kappa(W_d)$ and PER.

**Diagonal structure, Frobenius deviation, volume change, off-diagonal energy, orthogonality gap.** Figures 6, 7, 8, 9, and 10 demonstrate similar patterns: Card exhibits smoother, lower-magnitude metrics; Willett shows larger deviations, particularly in early days. All metrics correlate moderately with PER, further linking the learned linear transforms to recording quality.

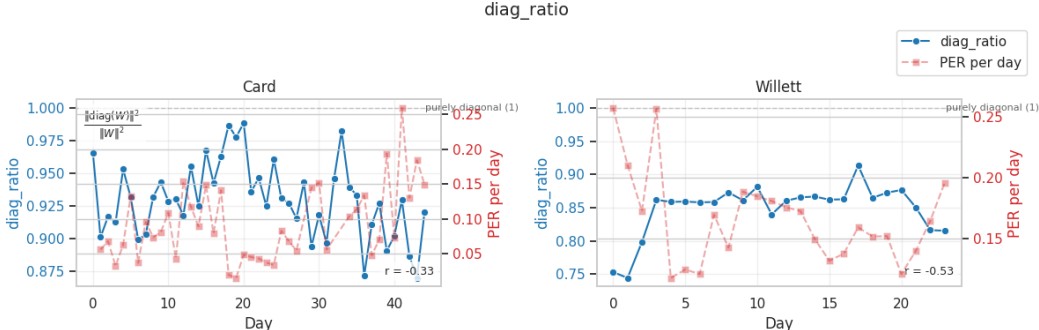

Figure 6: Day-wise evolution of diagonal ratio and PER.

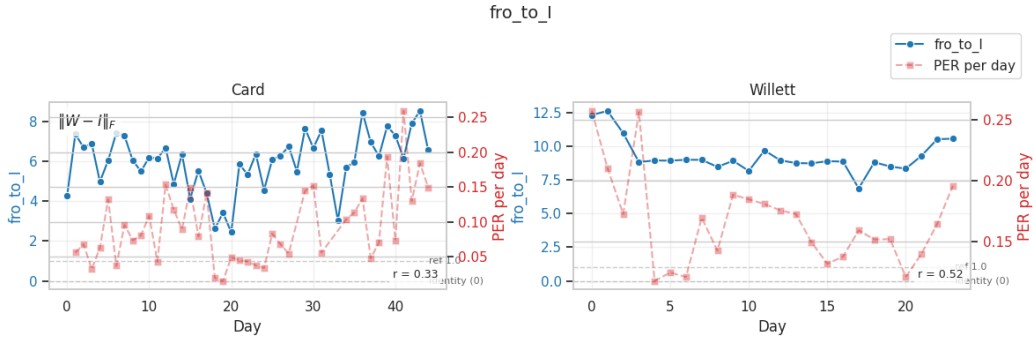

Figure 7: Day-wise evolution of $\|W_d - I\|_F$ and PER.

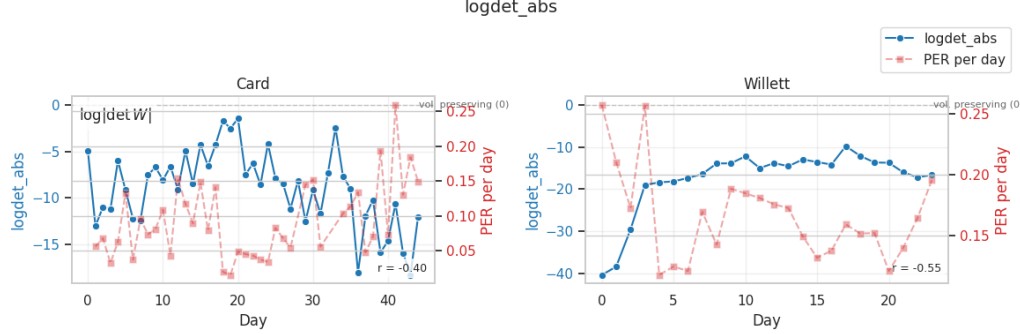

Figure 8: Day-wise evolution of $\log|\det W_d|$ and PER.

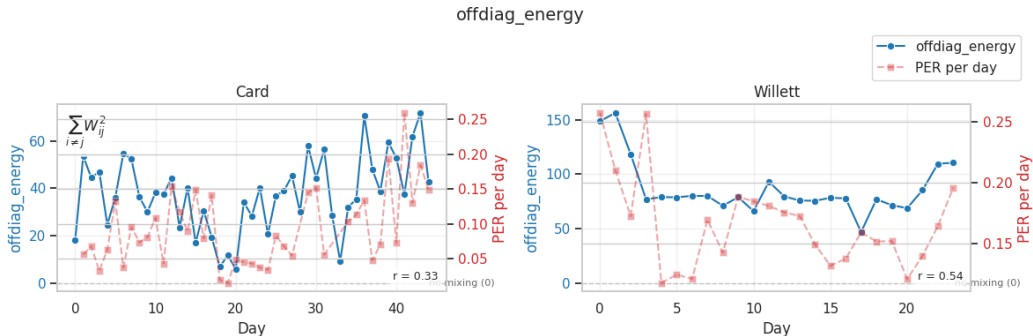

Figure 9: Day-wise evolution of off-diagonal energy and PER.

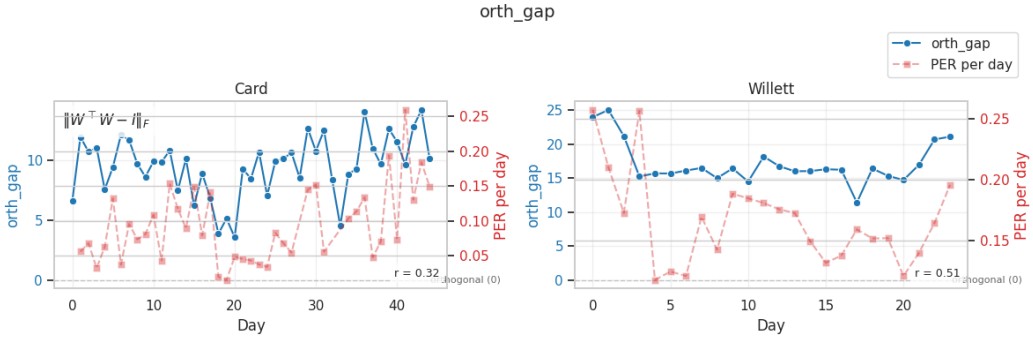

Figure 10: Day-wise evolution of orthogonality gap and PER.

## D  SENSITIVITY ANALYSES OF CROSS-DATASET TRAINING AND HIERARCHICAL CTC WEIGHT

To isolate the effect of key hyperparameters on cross-subject and cross-dataset performance, we conducted two controlled sensitivity analyses. All experiments were performed under identical training conditions, using a fixed random seed and identical optimization hyperparameters. Thus, each analysis varies exactly one factor at a time, enabling a clear causal interpretation of the results.

### D.1  EFFECT OF HIERARCHICAL CTC WEIGHT $\alpha$

The first analysis studies the contribution of intermediate CTC supervision by varying the loss weight $\alpha$ associated with the hierarchical CTC auxiliary head. The parameter $\alpha$ determines the relative influence of intermediate phoneme-level loss signals on the optimization dynamics. We sweep $\alpha \in \{0.1, 0.2, 0.3, 0.5, 0.75, 1.0\}$ while holding every other training component fixed.

**Numerical results.**  Table 5 reports the PER for both datasets. A distinct optimum emerges at $\alpha = 0.3$, which yields the lowest error for both Card and Willett.

Table 5: Sensitivity of PER to hierarchical CTC weight $\alpha$.

| $\alpha$ | Card PER | Willett PER |
|---|---|---|
| 0.1 | 9.4 | 16.5 |
| 0.2 | 9.5 | 17.1 |
| 0.3 | **9.1** | **16.1** |
| 0.5 | 10.1 | 17.6 |
| 0.75 | 10.6 | 18.5 |
| 1.0 | 10.7 | 18.7 |

**Interpretation.**  Moderate hierarchical supervision improves decoding, suggesting that intermediate phoneme representations are beneficial but should not dominate optimization. Both datasets exhibit the same optimum, which supports the idea that hierarchical CTC acts as a regularizer that stabilizes gradient flow and improves cross-subject phonetic structure.

Figure 11 visualizes the PER evolution with respect to $\alpha$, including baseline performance levels from single-dataset training. The vertical dashed line highlights the optimal configuration.

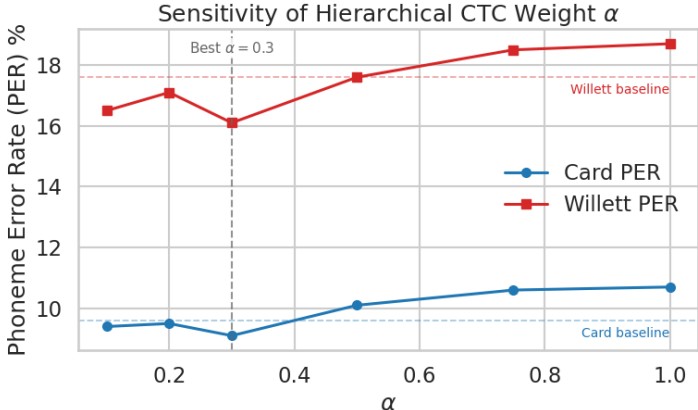

Figure 11: Sensitivity of PER to the hierarchical CTC weight $\alpha$. Dashed horizontal lines denote baselines; the best setting $\alpha = 0.3$ is marked by a vertical line.

## D.2 Cross-Dataset Fraction Sensitivity Analysis

The second analysis examines how the amount of data from a *second* dataset influences generalization. We consider two complementary configurations:

1. **Full Card + fraction of Willett**: measures how much Willett data is required to improve Willett decoding performance when starting from a strong Card-trained model.
2. **Full Willett + fraction of Card**: measures how Card data helps transfer performance onto Card subjects when starting from a Willett-trained model.

We further include "Card only" and "Willett only" controls, trained on the same fractions. As before, the random seed, optimizer, number of epochs, and model architecture remain fixed.

**Numerical results.** Tables 6 and 7 summarize the results.

Table 6: Willett PER as a function of the fraction of Willett data used.

| Fraction | Full Card + fraction Willett | Willett only |
|---|---|---|
| 10% | 43.1 | 47.8 |
| 25% | 31.2 | 36.6 |
| 50% | 24.5 | 30.7 |
| 75% | 19.3 | 26.4 |
| 100% | 16.1 | 19.7 |

Table 7: Card PER as a function of the fraction of Card data used.

| Fraction | Full Willett + fraction Card | Card only |
|---|---|---|
| 10% | 36.0 | 38.9 |
| 25% | 24.2 | 26.3 |
| 50% | 17.8 | 18.5 |
| 75% | 14.8 | 15.8 |
| 100% | 9.1 | 10.2 |

Our interpretation of these results is that adding even a small amount of data from the second dataset consistently improves performance relative to training on that dataset alone. This effect is present in both directions: Card improves Willett decoding, and Willett improves Card decoding. The improvement is especially pronounced at small fractions (10–25%), suggesting that cross-dataset pretraining provides strong structural priors for phoneme-level representations. At 100% fraction, both experiments converge to the same PER (16.1 for Willett, 9.1 for Card), matching the fully mixed training configuration.

Figure 12 shows the two learning curves side-by-side.

These experiments confirm that both hierarchical CTC supervision and cross-dataset exposure substantially improve generalization. The consistent optimum at $\alpha = 0.3$ demonstrates that hierarchical supervision provides beneficial intermediate gradients, while the fraction-sensitivity study illustrates a clear and reciprocal transfer effect between datasets. Both findings reinforce the central claim of the paper: **cross-subject and cross-dataset representations share a stable phonetic structure that can be leveraged through joint training and appropriate auxiliary losses**.

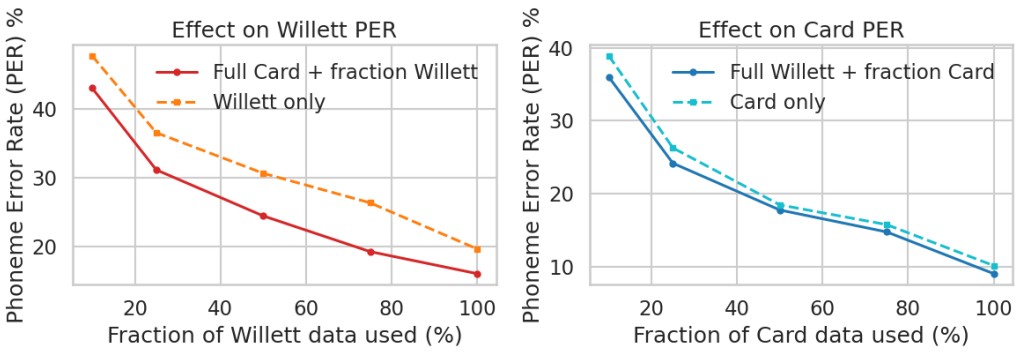

Figure 12: Sensitivity of PER to the fraction of the second dataset used. Left: Full Card + fraction Willett vs. Willett-only. Right: Full Willett + fraction Card vs. Card-only.

