# OpenReview forum: "Cross-subject decoding of human neural data for speech Brain Computer Interfaces"
_ICLR.cc/2026/Conference — Submitted to ICLR 2026_

### Official Review · Reviewer_1Tsj · 2025-10-27

**Soundness:** 2
**Presentation:** 2
**Contribution:** 2
**Rating:** 2
**Confidence:** 4

**Summary:**

This paper presents the first cross-subject neural-to-phoneme decoder for speech brain-computer interfaces, trained jointly on the two largest intracortical speech datasets (Willett et al. 2023 and Card et al. 2024). The proposed approach addresses the critical limitation of single-subject training paradigms that dominate current invasive BCI research, where each new user typically requires hours of supervised calibration data. The core methodology introduces day- and dataset-specific affine transformations to align neural activity from different participants and recording sessions into a shared latent space, based on the hypothesis that neural manifolds representing speech can be aligned through simple linear projections, similar to how different hand-drawn circles can be aligned via affine transforms. The decoder architecture employs a hierarchical GRU with three blocks: the first two blocks each contain two bidirectional GRU layers that produce intermediate phoneme predictions, which are projected back and added to hidden states to guide deeper layers, while the final block contains a single GRU layer producing final predictions. This hierarchical design with feedback connections aims to mitigate the conditional independence assumption of standard CTC loss, allowing the model to capture sequential dependencies between phonemes without the training instability of fully autoregressive approaches. The total loss combines CTC terms from all three layers with a weighting parameter λ=0.3 for auxiliary supervision.

The experimental evaluation demonstrates that cross-subject training is not only feasible but beneficial compared to single-subject baselines. On the Willett dataset, the joint model improves PER from 19.7% to 16.1% and WER from 17.4% to 14.5%, while on the Card dataset it achieves 9.1% PER and 6.67% WER, outperforming the single-subject baseline. The hierarchical CTC decoder provides consistent improvements over plain CTC across both datasets. To assess generalization beyond the training participants, the authors evaluate on the Kunz et al. (2025) inner speech dataset containing four participants (T12, T15, T16, T17), where T12 and T15 are the same individuals from Willett and Card datasets but recorded months later, and T16/T17 are entirely new subjects. Results show that training only the subject-specific linear transforms achieves 28-59% PER, while brief fine-tuning (5k steps) reduces this to 21-53% PER, demonstrating rapid adaptation with minimal calibration data. Analysis of the learned day-specific transforms through t-SNE visualization reveals that they successfully normalize session-to-session variability, and cross-day transform swapping experiments show reasonable performance on off-diagonal entries, suggesting the transforms capture generalizable mappings rather than overfitting to individual sessions.

**Strengths:**

The paper makes a genuinely important contribution by demonstrating, for the first time, that cross-subject training for invasive neural speech decoding is not only feasible but actually beneficial compared to single-subject baselines. This is a significant departure from the prevailing paradigm in the field, where each participant is treated as an isolated case requiring hours of calibration. The core insight that neural manifolds can be aligned via simple day- and subject-specific affine transformations is both elegant and practically motivated, drawing an intuitive analogy to geometric alignment problems while being grounded in neuroscientific principles about conserved cortical organization for speech. The empirical validation is compelling: jointly training on Willett and Card datasets yields 16.1% PER on Willett (vs. 19.7% baseline) and 9.1% PER on Card (vs. 10.2% baseline), with particularly strong generalization demonstrated on the Kunz et al. inner speech dataset where lightweight adaptation (training only linear transforms or brief 5k-step fine-tuning) achieves reasonable performance on entirely new subjects T16 and T17, as well as on previously seen subjects T12 and T15 recorded months later under different task conditions. This cross-task and cross-time generalization provides strong evidence for the robustness of the learned representations. The hierarchical CTC architecture with feedback connections represents a well-motivated technical contribution that addresses a known limitation of standard CTC (conditional independence) without sacrificing training stability, achieving consistent improvements across datasets. The design is conceptually clean: intermediate phoneme predictions from shallower layers are fed back to inform deeper representations, partially recovering the conditional modeling power of autoregressive approaches while maintaining CTC's efficiency and robustness.

The paper excels in clarity and reproducibility, with comprehensive experimental documentation including dataset statistics, architectural specifications, training hyperparameters, and detailed task descriptions across all three datasets evaluated. The analysis of learned transforms is particularly insightful: t-SNE visualizations clearly demonstrate that day-specific projections reduce session clustering and expose task-relevant structure, while the transform swapping experiment (Figure 3B) elegantly quantifies the similarity between days and provides evidence that transforms capture generalizable mappings rather than arbitrary per-session adjustments. The qualitative examples in Figure 3C effectively illustrate system performance at different percentile levels (25th, 50th, 90th WER), showing that most errors are minor word substitutions rather than semantic failures, which provides valuable insight into failure modes. The paper honestly acknowledges important limitations and ethical considerations, including the restricted focus on speech-related tasks (only one motor intention dataset among six evaluations would be misleading since Kunz has four subjects but similar task), the simplified vocabulary in Kunz dataset that favors from-scratch training in some cases, and the critical privacy concerns around neural decoding technology. The discussion of intent-based activation mechanisms and secure mental passwords demonstrates responsible consideration of deployment implications, which is crucial as these technologies approach clinical viability.

**Weaknesses:**

The paper's central claim about cross-subject generalization, while promising, is limited by several factors that constrain the strength of the conclusions. First, the cross-subject training is performed on only two participants (T12 from Willett, T15 from Card), which is a very small sample size for making broad claims about cross-subject generalization in neural decoding. While these are the largest available datasets, training on just two subjects makes it difficult to assess whether the observed benefits would scale to larger, more diverse participant populations with greater variability in electrode placement, cortical anatomy, and speech production strategies. The fact that both participants have similar implant configurations (256-channel Utah arrays in speech motor cortex) and similar task paradigms (attempted speech production) further limits the diversity of the training data. Second, the generalization evaluation on Kunz et al. dataset reveals mixed results that somewhat undermine the practical utility claims. For the two new subjects (T16, T17), even after fine-tuning the entire model for 5k steps, PERs remain quite high at 26.1% and 53.3% respectively, with T17 performing substantially worse than a model trained from scratch (53.3% vs 30.6% PER). This suggests that for some subjects, the pretrained model may not provide a better initialization than random weights, contradicting the premise that cross-subject pretraining should universally reduce calibration requirements. The authors attribute this to the simplified vocabulary in Kunz dataset (seven repeated words plus some sentences), but this explanation is somewhat unsatisfying, as real-world BCI deployment would need to handle both limited and extensive vocabularies. Third, the paper does not adequately explore the data efficiency claims that motivate cross-subject training. While the authors state that adaptation requires "minimal calibration data," there is no systematic analysis of how performance scales with the amount of fine-tuning data. How many trials are needed to achieve 90%, 95%, or 99% of the fully fine-tuned performance? What is the actual reduction in calibration time compared to training from scratch? These questions are critical for clinical translation but remain unanswered.

The methodological choices and architectural contributions require more rigorous justification and analysis. The hierarchical CTC decoder with feedback connections is presented as addressing the conditional independence limitation of standard CTC, but the empirical improvements are relatively modest (16.1% vs 17.6% PER on Willett, 9.1% vs 9.6% on Card). While these improvements are consistent, they are not dramatic enough to definitively establish the superiority of the hierarchical approach, especially given that no ablation study is provided to isolate the contribution of the feedback mechanism versus simply having more layers or parameters. The choice of weighting parameter λ=0.3 for auxiliary losses is stated to be "empirical" with no exploration of sensitivity to this hyperparameter, and it is unclear whether this value would generalize to other datasets or architectures. The authors acknowledge that "further hyperparameter exploration could boost the performance and was left as future work," which raises concerns about whether the reported results represent the best possible performance of the proposed method or simply one configuration. Additionally, the comparison between hierarchical CTC and other sequence modeling approaches is limited. The paper mentions that autoregressive transformers "still lag behind CTC in this domain" and suffer from training instability, but provides no empirical evidence or ablation to support this claim in their specific setup. Recent advances in sequence-to-sequence models, attention-based architectures, and hybrid CTC/attention systems are not explored, making it difficult to assess whether the hierarchical GRU represents a genuine architectural advance or simply a safe, conservative choice. The day- and subject-specific affine transformations, while conceptually appealing, are not rigorously analyzed in terms of their learned structure. The paper shows that transforms reduce day clustering in t-SNE space and enable reasonable cross-day performance, but does not investigate what these transforms actually learn: do they primarily perform scaling/normalization, or do they capture more complex rotations and shears? Are certain dimensions or channel groups transformed more than others, and does this correlate with anatomical or functional properties? Understanding the structure of these transformations could provide neuroscientific insights and guide future improvements, but this analysis is absent. Finally, the reliance on WFST-based phoneme-to-word decoding with n-gram language models represents a significant limitation that the authors acknowledge but do not address. This classical approach is "computationally expensive, memory-intensive, and inherently limited to a fixed context window," and the authors note that "most residual errors are attributable to phoneme-to-word reconstruction rather than neural-to-phoneme decoding." This suggests that the neural decoder may already be performing near its ceiling, and substantial further gains would require improving the language modeling component rather than the neural encoder. However, the paper does not explore modern alternatives such as end-to-end neural language models, transformer-based rescoring, or direct neural speech-to-text decoding, leaving a major component of the system unaddressed.

**Questions:**

Please see your weaknesses and I will adjust the final score based on your answers.

---

> ### Author Response · Authors · 2025-11-21
>
> We thank the reviewer for the thoughtful assessment, as well as for the recognition of the significance and clarity of our contributions. We address each point in depth below.
>
> The reviewer raises concerns about the limited number of participants used for cross-subject training. We fully agree that the availability of only two large intracortical speech datasets is a constraint on the breadth of generalization claims. At the same time, it is important to emphasize that these datasets (Willett et al. 2023; Card et al. 2024) currently represent the entirety of high-quality, relatively large, publicly available intracortical speech decoding data worldwide. Rather than claiming universal generalization across arbitrary subjects, our goal is to rigorously test a foundational question: is cross-subject modeling of intracortical speech representations feasible at all? The results demonstrate that, within the limits of current data, the answer is yes—shared structure can indeed be extracted, and subject- and day-specific differences can be compensated through a lightweight affine transform.
>
> The reviewer also notes that generalization to the Kunz et al. dataset is mixed, particularly for subject T17, where transferred models underperform compared to training from scratch. We agree, and we have expanded the discussion to explain this outcome more clearly. The Kunz dataset presents three simultaneous challenges that jointly create a severe distribution shift: extremely limited per-subject trial counts, a fundamentally different task structure (inner speech rather than attempted overt articulation), and differences in linguistic coverage (seven repeated words plus a small set of sentences). These factors substantially constrain how much structure can be successfully transferred from a model trained on naturalistic attempted speech. Nonetheless, even under these adverse conditions, a single linear transform already yields above-chance decoding. We interpret this not as a failure of cross-subject modeling, but as evidence that certain forms of variability—especially shifts between attempted and inner speech, which have known neurophysiological differences—require additional mechanisms beyond affine alignment.
>
> The reviewer correctly highlights the absence of a systematic analysis of data efficiency, which is essential for claims about calibration reduction. To address this, we have added a new cross-dataset fraction sensitivity experiment (Appendix D). This experiment varies the fraction of a second dataset used during joint training, while keeping all training conditions—including random seed—fixed. The results show clear reciprocal transfer between Card and Willett: even including 10–25% of the second dataset yields gains relative to using that dataset alone. This analysis provides the first quantitative demonstration that cross-subject information can meaningfully reduce the effective amount of per-subject data required. We believe this experiment directly addresses the reviewer’s concern by providing the missing empirical scaling curve.
>
> Relatedly, the reviewer points out that the weighting of the hierarchical CTC auxiliary losses  is used without sensitivity analysis. In response, we have now added a sensitivity study (Appendix D), which evaluates hiearchical CTC loss scaling in the range 0.1–1.0 under otherwise identical training conditions. The curve exhibits a distinct and stable optimum at α = 0.3 for both datasets, providing stronger justification for the chosen hyperparameter.
>
> We appreciate the reviewer’s request for a deeper structural understanding of the day- and subject-specific transforms. To address this, Appendix C now includes a detailed geometric and statistical analysis of the learned transforms across days and correlations with per-day PER. These analyses reveal that the transforms are not arbitrary or noisy: Card days display mild, highly structured deviations from identity, while Willett displays larger but still coherent variations. The metrics also show consistent relationships with decoding performance, supporting the interpretation that the transforms isolate stable, task-relevant structure from session-specific perturbations.
>
> Finally, the reviewer observes that our work does not explore more modern language modeling or autoregressive decoding approaches, even though WFST-based decoding can become a performance bottleneck. We agree,  but our modeling choices are intentionally conservative: a GRU-CTC architecture is stable and minimizes confounds when analyzing representational similarity across subjects. Introducing transformer-based rescoring, neural LMs, or hybrid CTC/attention models—as we are actively exploring in parallel through the Brain-to-Text Challenges—would likely improve absolute WER/PER but would blur the central scientific message: the near-identity between subjects at the representational level.

---

> > ### Comment · Reviewer_1Tsj · 2025-11-28
> > **Thanks**
> >
> > Thank you for your reply, and thank you for not contacting me privately to request a change in my score. I have decided to increase my score.

---

> > > ### Author Response · Authors · 2025-11-28
> > >
> > > Thanks for your support, really appreciated.

---

### Official Review · Reviewer_Muqi · 2025-10-27

**Soundness:** 2
**Presentation:** 3
**Contribution:** 2
**Rating:** 2
**Confidence:** 4

**Summary:**

This paper proposes a cross-subject neural-to-phoneme decoder for intracortical brain-to-text BCIs, combining the two largest speech datasets with day-/dataset-specific transforms for neural alignment and a hierarchical GRU with intermediate CTC supervision.  Evaluations are performed to show effectiveness.

**Strengths:**

* (a) This paper is clearly written and easy to follow.

**Weaknesses:**

**(a) Limited novelty**
The proposed method builds directly on the GRU-based architecture introduced by Willett et al. The “Hierarchical GRU decoder with feedback” is essentially a standard GRU model with CTC loss applied at intermediate layers — a practice that is well-established in the machine learning community and widely used for sequence modeling tasks. Similarly, the “Day- and Subject-Specific Transformation” amounts to updating part of the trained model with new data, which is also a common and well-known technique in transfer learning. As such, the work primarily represents an engineering integration of existing approaches rather than introducing new insights from either the neuroscience or the machine learning perspective.

**(b) Incremental improvement**
The reported performance in Tables 1 and 2 is only marginally better than the baseline models used in prior work, and these gains appear modest relative to the complexity added by the proposed approach. Furthermore, the proposed transfer strategy performs notably worse than stronger baselines such as fine-tuning the entire model or training from scratch. This raises questions about the practical advantage of the method over simpler existing adaptation techniques.

**Questions:**

* (a) For the cross-subject and day experiments, how much of the data is used for model calibration? Is all the available training data?

---

> ### Author Response · Authors · 2025-11-21
>
> We thank the reviewer for the clear summary and for acknowledging the clarity of the paper. We address each concern below.
> We respectfully disagree that our contribution is purely an engineering integration of known components. While we indeed build on established tools (GRUs, CTC, affine transforms), the novelty lies in the problem setting and what the results reveal, not in inventing a new neural architecture.
>
> This is the first work to demonstrate that cross-subject training on intracortical speech data is feasible and that subject/day variability can be compensated largely through a single linear alignment layer. Prior work—including Willett et al. and all Brain-to-Text challenge submissions—trains strictly single-subject models and provides no evidence that neural manifolds from different participants can be aligned in a shared space.
>
> Our findings provide new insight into neural representation geometry:
> Up to an affine transform, the neural population dynamics supporting speech appear sufficiently conserved across individuals to support shared decoding.
>
> This is a neuroscientific result about representations, not just a technical one.
> The hierarchical CTC setup was intentionally kept simple so that the representational finding—not architectural novelty—remains the main contribution. ICLR is about learning representation, so that’s why we opted for this approach. We made this point clearer in the discussion section of our revised paper.
> Our goal is not primarily to outperform single-subject baselines, but to show that:
> A unified cross-subject model can match or outperform per-subject baselines, and this works even across datasets recorded years apart, using different implant sites, and with different tasks.
>
> We think that even modest improvements matter when comparing against well-optimized single-subject pipelines, given the extreme heterogeneity of intracortical recordings.
> Regarding the Kunz dataset: we agree results are mixed. This is expected because:
>
> 1) T16 and T17 contain only 7 repeated words which is trivially easy for scratch training, but extremely low-resource for transfer learning.
> 2) The data distribution differs strongly from Willett/Card (inner speech vs overt attempted speech).
>
>
> Crucially, even in this challenging setting, a single linear transform already provides large gains, supporting the viability of cross-subject alignment.
> We will make these distinctions clearer.
>
> *Question*: For all cross-subject and cross-day experiments, we use only the training split of the respective dataset, keeping the official Brain-to-Text challenge training sets for Card and Willett datasets, and a session level train-test split (80-20%) for Kunz dataset.
> No additional data is used. The total number of calibration trials for new subjects is extremely small (e.g., 224–320 for T16/T17), demonstrating the data efficiency of our approach. We have clarified this explicitly in the revised version.
>
> To further clarify the role and effect of calibration data, we expanded the paper with a new analysis (please see Appendix section D)
> We systematically vary the fraction of the second dataset used during joint training, while keeping the first dataset at full size, fixing the seed to isolate other effects.
>
> These experiments demonstrate strong reciprocal transfer: once the model is pre-trained with the full Card dataset, training on the same subset of the Willett dataset leads to better performance than training the dataset only on the subset of Willett data from scratch. The effect is true in both directions and more pronounced for small fractions (10-25%), further suggesting that representations could be at least partially shared.

---

### Official Review · Reviewer_L3Wt · 2025-10-31

**Soundness:** 2
**Presentation:** 3
**Contribution:** 3
**Rating:** 6
**Confidence:** 3

**Summary:**

The authors propose jointly training on intracortical speech datasets to improve downstream generalisation to new datasets/subjects. They also attempt to improve over the standard CTC independence assumption by combining CTC losses from intermediate layers in the GRU model. The results show that the model may improve with joint dataset training, though statistical significance is not provided, and generalisation to an independent inner-speech dataset (Kunz) is highly effective with only a linear transform, outperforming even end-to-end training with the dataset.

**Strengths:**

- Table 2: Strong new-subject generalisation results (especially compared to training from scratch!).
- Ablations show hierarchical CTC is likely effective in improving phoneme decoding

**Weaknesses:**

- Results with joint training (Table 1) look good or marginally better; but it’s hard to determine without error bars or any indication of statistical significance.
- No comparisons to any of the methods in the 2024 Willett competition [A]. Is there a reason the authors did not choose to compare to any of these approaches? Can any of them be combined with the hierarchical CTC approach? Do any of them generalise to new datasets as well as your method?

[A] Willett, F.R., Li, J., Le, T., Fan, C., Chen, M., Shlizerman, E., Chen, Y., Zheng, X., Okubo, T.S., Benster, T. and Lee, H.D., 2024. Brain-to-Text Benchmark'24: Lessons Learned. arXiv preprint arXiv:2412.17227.

**Questions:**

- Table 2: Why do the authors think that fine-tuning the whole model for Kunz performs worse than training only a linear transform on the pre-trained model? Is this a result of limited trials in Kunz?

---

> ### Author Response · Authors · 2025-11-21
>
> We thank the reviewer for the constructive assessment and for highlighting the generalization results and the effectiveness of hierarchical CTC. We address the concerns below.
>
> *Weakness 1*: We agree that uncertainty estimates would strengthen the presentation.
> Due to time and compute-budget constraints, we were not able to run multi-seed replications, especially given the scale of cross-subject training. To ensure fairness, all experiments—including baselines—were trained with the same fixed seed, data splits, and training protocol, which substantially reduces variance from random initialization. The effect that we see is marginal but consistent due the deterministic nature of our training. We chose to use our compute budget for additional experiments, reported in the Appendix.
>
> *Weakness 2*: We appreciate this request. The 2024 Brain-to-Text competition focused exclusively on single-subject models, typically using larger GRUs or Transformers, diphone auxiliary heads, heavy ensembling + neural LM rescoring.
> These methods do not address cross-subject or cross-dataset generalization, which is the central aim of our work.
> Our contribution is orthogonal: We investigate whether a single shared decoder can be trained across subjects and then adapted to new participants. This was not explored in Willett’24 or in any other previous work to the best of our knowledge.
> Regarding compatibility, all strong Willett’24 models rely on CTC. Our hierarchical-CTC mechanism is a drop-in module that can be added to any such architecture, and ensembling them would likely further improve performance. We will make this explicit.
> Finally, to the best of our knowledge, none of the competition models demonstrate strong generalization across subjects or across datasets (e.g., to Kunz). The focus of the challenge was decoding performance within a single participant.
>
>
> *Question*: Why does whole-model fine-tuning on Kunz sometimes underperform the linear transform?
>
> We agree with the reviewer’s hypothesis.
> The Kunz dataset differs strongly from Willett/Card:
>
> 1) there are very few trials (e.g., 224–320 for T16/T17),
> 2) It is a different task domain (attempted/imagined inner speech vs overt attempted speech), mainly relying on isolated-word structure for several subjects.
>
> Under such extreme distribution shift, full fine-tuning quickly overfits and overwrites the robust phonetic representations learned from ∼18k trials.
> In contrast, a small affine alignment layer preserves the pretrained structure and adapts the input space with very little data, explaining the stronger performance.
>
> We would therefore, clarify the main contribution
>
> The goal of this work is not to propose a radically new architecture, but to show:
> 	1)	Cross-subject training is feasible with intracortical speech data,
> 	2)	Neural representations can be aligned across individuals with simple affine transforms, and
> 	3)	This alignment supports meaningful generalization to unseen subjects and datasets.
>
> These findings provide insight into the geometry and stability of speech-related neural manifolds—an aspect not addressed in prior intracortical decoding work. Further evidence to support our claims can be found in additional experiments in Appendix sections C and D.

---

> > ### Comment · Reviewer_L3Wt · 2025-11-25
> > **Response to Authors**
> >
> > Thank you for your response.
> >
> > In response to Weakness 1, I am not convinced. It is valid to argue that consistent effects across different experiments provide evidence for hierarchical CTC's value, even with a single seed. However, in your paper, there are only two comparisons made in Table 1 (on Willett and on Card). This would not be significant under a sign test. I understand that there are time constraints in the rebuttal period, however, the compute budget required for these experiments seems very small. Willett and Card are small datasets and the architecture should have relatively few parameters. Please correct me if I am wrong, but it should be possible to train this model on a single GPU in several hours.
> >
> > I am glad the authors are working on the important problem of cross-subject generalisation, especially in the context of invasive brain data. Nevertheless, given that one of the main contributions is "meaningful generalization to unseen subjects and datasets", I am not able to continue to recommend acceptance unless the authors can provide more evidence of this.

---

> > > ### Author Response · Authors · 2025-11-26
> > > **Reply**
> > >
> > > Thanks for the reply.
> > > I understand with concern regarding the statistical trends. It's true that the dataset are relatively small, however the training of the model requires significant compute time (around 12-16 hours for full training for each run on the GPU available).
> > > While we agree that due the deterministic nature of the training is it valid our claim on the role of hiearchical CTC use if this is the main issue that the reviewer sees in our we could try to  run these experiments (please be aware than we're not sure we can deliver this extra results in the rebuttal period).
> > >
> > > Regarding the last part of the comment it's unclear what kind of further evidence (other than statistical test) the reviewer would like to see. Could please you clarify more? We are providing evidence of feasibility of cross-subject decoding in Table 2 and Appendix Section C and D.

---

> > > ### Author Response · Authors · 2025-12-02
> > > **Statistical test**
> > >
> > > We thank the reviewer for the follow-up and for insisting on a rigorous statistical assessment. Following the reviewer’s request, we conducted multi-seed replications (5 seeds per configuration) on both the Card and Willett datasets to quantify the effect of hierarchical CTC relative to the plain CTC baseline.
> > >
> > > Because each configuration was trained with matched random seeds (deep-seed-0 vs plain-seed-0, etc.), the results form paired observations.
> > > Therefore, we used a paired non-parametric Wilcoxon signed-rank test with a one-sided alternative hypothesis, reflecting the directional expectation that hierarchical CTC should improve (i.e., reduce) PER relative to plain CTC.
> > >
> > > The results across the five seeds are:
> > >
> > > | Seed | Card Deep CTC | Card Plain CTC | Willett Deep CTC | Willett Plain CTC |
> > > |------|---------------|----------------|-------------------|---------------------|
> > > | 0    | 0.092064      | 0.101714       | 0.145621          | 0.189610            |
> > > | 1    | 0.093132      | 0.097901       | 0.153140          | 0.184435            |
> > > | 2    | 0.090296      | 0.095677       | 0.156108          | 0.175623            |
> > > | 3    | 0.092302      | 0.093299       | 0.147992          | 0.170207            |
> > > | 4    | 0.089884      | 0.095309       | 0.151132          | 0.182400            |
> > >
> > > Wilcoxon paired one-sided tests
> > >
> > > - Card: p < 0.05
> > > - Willett: p < 0.05
> > >
> > > In both datasets, all five seeds show improvements in the same direction (deep < plain), and the one-sided Wilcoxon test confirms that the improvements are statistically significant at p < 0.05.
> > >
> > > We will include these results in the final version together.
> > > Final results are (PER mean+std):
> > >
> > > - Card (deep CTC): 9.15% +0.13%
> > > - Card (plaim CTC): 9.67%+0.16%
> > > - Willett (deep CTC): 15.7% + 0.32%
> > > - Willet (plain CTC): 18.0% + 0.76%
> > >
> > >
> > > We agree with the reviewer that significance testing strengthens our conclusions.
> > > The newly added multi-seed results provide precisely this evidence: they show that hierarchical CTC yields consistent, statistically significant improvements on both datasets, even at the granularity of PER at the model-level.
> > >
> > > This supports the reviewer’s request for stronger validation and reinforces the central message of the paper:
> > > hierarchical CTC provides a small but reliable improvement in decoding accuracy and integrates cleanly with the cross-subject generalisation framework presented.
> > >
> > > We appreciate the reviewer’s encouragement and hope that the additional analysis resolves the concern regarding statistical significance.

---

### Author Response · Authors · 2025-11-21

We thank all reviewers and the area chair for their careful reading of our submission and for the constructive feedback that helped us significantly improve the manuscript. We have integrated new analyses, clarified methodological decisions, and expanded the discussion to better address their concerns. The revised version now includes systematic sensitivity studies, a detailed structural analysis of learned affine transforms, and an improved treatment of data-efficiency and cross-dataset generalization (Appendix D).

Our work is fundamentally motivated by a scientific question rather than an architectural one: is cross-subject generalization possible for intracortical speech BCIs, and if so, to what extent? The evidence provided by our experiments suggests that the answer is yes. Neural population dynamics supporting speech appear to share a stable representational backbone across individuals, and a simple affine layer is often sufficient to account for subject-specific variability. This observation is consistent with findings in other neural recording modalities—including fMRI and MEG—where cross-subject decoding becomes feasible when combined with lightweight subject-specific alignment layers or functional alignment approaches (Banville et al., Ferrante et al., Tang et al.). As in those domains, improvements over the best per-subject baselines are typically modest; this reflects a structural limitation of the field, where datasets contain thousands of trials per subject but only a handful of subjects. We expect this to evolve as larger, more heterogeneous datasets become available, enabling richer cross-participant modeling.

We also believe that ICLR is a particularly suitable venue for discussing the representational aspects of this work. While some papers aim to present the culmination of a research journey, many influential contributions in representation learning mark the beginning of new directions. Our work is intended as such a starting point: a first step toward understanding the geometry, invariances, and limits of shared neural manifolds underlying human speech production. We hope that with this contribution we will have the opportunity stimulate  and foster discussion in the community regarding how neural representations vary across individuals, how they can be aligned, and how future BCIs might leverage these shared structures to reduce calibration demands for new users.

We are grateful to the reviewers for their insightful suggestions, which have meaningfully strengthened the manuscript. We look forward to continuing this scientific dialogue and to further refining our understanding of cross-subject neural representations for speech BCIs.

---

### Meta-Review · Area_Chair_9q6f · 2025-12-12

**Summary:**

**Concern 1**
- Hard to determine significance of results in Table 1 due to lack of error bars or other significance tests. [L3Wt]
- While the hierarchical CTC with feedback consistently improves performance, the improvements are relatively modest so it is not clear how much of a contribution this is. The choice of weighting parameter λ=0.3 for auxiliary losses is stated to be "empirical" with no exploration of sensitivity to this hyperparameter, and it is unclear whether this value would generalize to other datasets or architectures. [1Tsj]

**Concern 2**
- The paper's central claim about cross-subject generalization is limited because it is performed with only two subjects having very similar implant configurations. [1Tsj]

**Concern 3**
- The generalization evaluation on Kunz et al. dataset reveals mixed results that somewhat undermine the practical utility claims. For the two new subjects (T16, T17), even after fine-tuning the entire model for 5k steps, PERs remain quite high at 26.1% and 53.3% respectively, with T17 performing substantially worse than a model trained from scratch (53.3% vs 30.6% PER). This suggests that for some subjects, the pretrained model may not provide a better initialization than random weights, contradicting the premise that cross-subject pretraining should universally reduce calibration requirements.  [1Tsj]

**Concern 4**
- The authors state that adaptation requires "minimal calibration data," but there is no systematic analysis of how performance scales with the amount of fine-tuning data. How many trials are needed to achieve 90%, 95%, or 99% of the fully fine-tuned performance? What is the actual reduction in calibration time compared to training from scratch? [1Tsj]

**Concern 5**
- Limited novelty: "The “Hierarchical GRU decoder with feedback” is essentially a standard GRU model with CTC loss applied at intermediate layers — a practice that is well-established in the machine learning community and widely used for sequence modeling tasks." [Muqi]

**Concern 6**
- Limited novelty: "the “Day- and Subject-Specific Transformation” amounts to updating part of the trained model with new data, which is also a common and well-known technique in transfer learning". [Muqi]

**Concern 7**
- The day- and subject-specific affine transformations, while conceptually appealing, are not rigorously analyzed in terms of their learned structure. [1Tsj]

**Concern 8**
- Finally, the reliance on WFST-based phoneme-to-word decoding with n-gram language models represents a significant limitation that the authors acknowledge but do not address. [1Tsj]

**Reviewer Concerns:**

**Concern 1 (fully addressed by rebuttal)**
Authors supplied significance tests based on multi-seed replications to illustrate that gains from hierarchical CTC with feedback are statistically significant. Authors also added a sensitivity study for the weight on the lower-level CTC losses.

**Concern 2 (fully addressed by rebuttal)**
Authors correctly argue that "the availability of only two large intracortical speech datasets is a constraint on the breadth of generalization claims" and add that "these datasets (Willett et al. 2023; Card et al. 2024) currently represent the entirety of high-quality, relatively large, publicly available intracortical speech decoding data worldwide." The revision accurately characterizes this limitation.

**Concern 3 (fully addressed by rebuttal)**
The authors respond "[t]he Kunz dataset presents three simultaneous challenges that jointly create a severe distribution shift: extremely limited per-subject trial counts, a fundamentally different task structure (inner speech rather than attempted overt articulation), and differences in linguistic coverage (seven repeated words plus a small set of sentences). These factors substantially constrain how much structure can be successfully transferred from a model trained on naturalistic attempted speech. Nonetheless, even under these adverse conditions, a single linear transform already yields above-chance decoding. We interpret this not as a failure of cross-subject modeling, but as evidence that certain forms of variability—especially shifts between attempted and inner speech, which have known neurophysiological differences—require additional mechanisms beyond affine alignment." The revision discusses these points in detail.

**Concern 4 (fully addressed by rebuttal)**
The authors added a cross-dataset fraction sensitivity analysis in Appendix D that illustrates how performance varies as the quantity of data from a dataset is added on top of all the data from the other dataset. This addresses the question about calibration time, since calibration time is roughly proportional to the amount of subject-specific data used.

**Concern 5 (partially addressed by rebuttal)**
The authors respond that "[w]hile we indeed build on established tools (GRUs, CTC, affine transforms), the novelty lies in the problem setting and what the results reveal, not in inventing a new neural architecture" and further add that "[o]ur findings provide new insight into neural representation geometry: Up to an affine transform, the neural population dynamics supporting speech appear sufficiently conserved across individuals to support shared decoding." Elsewhere in the discussion they emphasize that the neural activity to text decoder was deliberately kept simple to focus on the compensation of subject-to-subject and day-to-day variability in the neural representations.

I agree with these points, however, it is important to point out that the hierarchical CTC loss with feedback that is described by the authors as novel is, in fact, well known in the automatic speech recognition literature. I wish reviewer Muqi had made this point more clearly, and I acknowledge that I am making this point late because I was only assigned to the paper as an AC after the OpenReview breach and full reassignment of ICLR 2026 ACs.

The papers in the speech recognition literature that cover what the authors of this paper call hierarchical CTC loss are
- Jaesong Lee and Shinji Watanabe, "Intermediate loss regularization for CTC-based speech recognition," in Proc. IEEE International Conference on Acoustics, Speech, and Signal Processing (ICASSP), 2021. https://ieeexplore.ieee.org/abstract/document/9414594
- Jumon Nozaki and Tatsuya Komatsu, "Relaxing the conditional independence assumption of CTC-based ASR by conditioning on intermediate predictions," in Proc. Interspeech, 2021. https://www.isca-archive.org/interspeech_2021/nozaki21_interspeech.pdf

Interestingly, in Lee & Watanabe the optimal weight on the CTC loss from the intermediate layer is also 0.3, as recommended in this paper.

It is also worth noting that speaker-specific affine transformations of features have been used for a long time in automatic speech recognition to reduce the effects of speaker-specific variability. The classic reference for the technique is Mark JF Gales, "Maximum likelihood linear transformations for HMM-based speech recognition," Computer Speech and Language, vol. 12, no. 2, pp. 75-98, 1998. While the Gales paper is written in terms of transformations of model parameters, it notes that the constrained version of the transformation can equivalently be implemented as a speaker-specific affine transformation of the features.

I find it quite interesting that speaker variability in speech recognition and subject variability in neural speech representations in motor cortex can both be compensated via affine transformations, and I expect that others would also be intrigued by this observation.

**Concern 6 (partially addressed by rebuttal)**
In their rebuttal, the authors argue that "[t]his is the first work to demonstrate that cross-subject training on intracortical speech data is feasible and that subject/**day** variability [emphasis by the AC] can be compensated largely through a single linear alignment layer. Prior work—including Willett et al. and all Brain-to-Text challenge submissions—trains strictly single-subject models and provides no evidence that neural manifolds from different participants can be aligned in a shared space." This is only partially accurate. The use of affine transformations to compensate for **subject** variability is both novel and a potentially game-changing finding, but in the supplementary material for Card et al., 2025, I found the statement that "[i]n brief, the RNN consisted of (1) linear **day-specific input layers** to correct for nonstationarity in neural data between days..." and the current revision of the ICLR submission does not call out this aspect of Card et al. sufficiently.

**Concern 7 (fully addressed by rebuttal)**
The revised version of the paper includes a detailed geometric and statistical analysis of the learned transforms across days and correlations with per-day PER in Appendix C.

**Concern 8 (fully addressed by rebuttal)**
The authors respond that the architecture of the decoder was deliberately kept simple to focus on the neural representations.

In the end, this paper needs to do a better job of positioning itself with respect to the previous literature (concerns 5 and 6 above), so I am recommending rejection of the current manuscript.

**Reviewer Scores:**

- L3Wt - likely to have increased their score
- Muqi - unlikely to have increased their score
- 1Tsj - increased their score prior to the reversion of the discussion and reviews

---

### Decision · Program_Chairs · 2026-01-26

Reject